# Hydroethanolic Extract of *Lepidium apetalum* Willdenow Alleviates Dextran Sulfate Sodium-Induced Colitis by Enhancing Intestinal Barrier Integrity and Inhibiting Oxidative Stress and Inflammasome Activation

**DOI:** 10.3390/antiox13070795

**Published:** 2024-06-29

**Authors:** Kwang-Youn Kim, Yun-Mi Kang, Ami Lee, Yeon-Ji Kim, Kyung-Ho Kim, Youn-Hwan Hwang

**Affiliations:** 1Korean Medicine (KM)-Application Center, Korea Institute of Oriental Medicine (KIOM), Daegu 41062, Republic of Korea; ymkang1013@kiom.re.kr (Y.-M.K.); yjikim@kiom.re.kr (Y.-J.K.); jk6012@kiom.re.kr (K.-H.K.); 2Herbal Medicine Research Division, Korea Institution of Oriental Medicine (KIOM), Daejeon 34054, Republic of Korea; dmb01367@kiom.re.kr; 3Korean Convergence Medical Science Major, KIOM School, University of Science & Technology (UST), Daejeon 34054, Republic of Korea

**Keywords:** *Lepidium apetalum* willdenow, ulcerative colitis, tight junction, oxidative stress, inflammasome

## Abstract

The prevalence of ulcerative colitis (UC) has surged in Asian nations recently. The limitations of traditional drug treatments, including biologics, have spurred interest in herbal medicines for managing UC. This study aimed to elucidate the protective mechanisms of hydroethanolic extract from Lepidium apetalum Willdenow (LWE) on intestinal integrity and inflammation in a dextran sodium sulfate (DSS)-induced colitis model of inflammatory bowel disease (IBD). Using UPLC-MS/MS analysis, eleven phytochemicals were identified in LWE, including catechin, vicenin-2, and quercetin. LWE restored transepithelial electrical resistance (TEER) and reduced paracellular permeability in IL-6-stimulated Caco-2 cells, increasing the expression of the tight junction proteins ZO-1 and occludin. LWE treatment alleviated DSS-induced colitis symptoms in mice, reducing body weight loss, disease activity index values, and spleen size, while improving colon length and reducing serum FITC-dextran levels, indicating enhanced intestinal barrier function. LWE suppressed NLRP3 inflammasome activation, reducing protein levels of pro-caspase-1, cleaved-caspase-1, ASC, and NLRP3, as well as mRNA levels of IL-1β, IL-6, and TNF-α. LWE treatment upregulated activity and mRNA levels of the antioxidant enzymes SOD1 and NQO1. Additionally, LWE modulated the Nrf2/Keap1 pathway, increasing p-Nrf2 levels and decreasing Keap1 levels. LWE also restored goblet cell numbers and reduced fibrosis in DSS-induced chronic colitis mice, increasing gene and protein expressions of ZO-1 and occludin. In summary, LWE shows promise as a therapeutic intervention for reducing tissue damage and inflammation by enhancing intestinal barrier function and inhibiting colonic oxidative stress-induced inflammasome activation.

## 1. Introduction

Inflammatory bowel disease (IBD) encompasses chronic inflammatory conditions affecting the gastrointestinal tract, including ulcerative colitis (UC) and Crohn’s disease (CD) [1]. Hallmarks of IBD include chronic inflammation and disruptions in intestinal barrier function [2]. Both UC and CD exhibit periods of active disease (flares) alternating with periods of remission, leading to various symptoms and complications [3]. The exact cause of IBD remains unclear, but is believed to involve a combination of genetic, environmental, and immune system factors. Treatment aims to reduce inflammation, alleviate symptoms, and maintain remission through medications such as anti-inflammatory drugs, immunosuppressants, biologics, and corticosteroids [4,5]. Lifestyle modifications, dietary changes, and surgery may also be part of the treatment plan, tailored to the individual’s specific needs and the disease severity [6,7].

In IBD, chronic inflammation in the gastrointestinal tract disrupts the structure and function of tight junctions, increasing intestinal permeability [8,9]. This allows luminal contents like bacteria, toxins, and food antigens to leak into underlying tissues, triggering immune responses and further inflammation [10,11]. Dysfunctional tight junctions perpetuate inflammation by allowing luminal antigens to cross the intestinal barrier and interact with immune cells in the gut mucosa [12,13]. This interaction activates immune pathways, releases pro-inflammatory cytokines, and recruits inflammatory cells, exacerbating tissue damage and inflammation. Studies have shown correlations between tight junction dysfunction and the severity, activity, and relapse of IBD in patients. Consequently, restoring tight junction integrity and reducing intestinal permeability have emerged as potential therapeutic strategies for managing IBD [14]. Pharmacological agents targeting tight junction proteins or modulating tight junction function are being explored in preclinical and clinical studies as potential treatments for IBD [15].

Pyroptosis is a type of programmed cell death that is associated with inflammation and mediated by the activation of inflammasomes [16]. In response to various stimuli, such as microbial pathogens, damage-associated molecular patterns, or cytokines, inflammasomes become activated in immune cells, particularly macrophages and dendritic cells, within the intestinal mucosa [17]. These multiprotein complexes play a central role in initiating the inflammatory response. Upon activation, inflammasomes trigger caspase-1, a protease enzyme involved in initiating pyroptosis and promoting the maturation and secretion of pro-inflammatory cytokines, particularly interleukin (IL)-1β and IL-18 [18,19]. Dysregulated inflammasome activation and pyroptosis have been implicated in the pathogenesis and progression of IBD [20,21]. Excessive or dysregulated production of IL-1β and IL-18, along with other pro-inflammatory mediators released during pyroptosis, can contribute to mucosal inflammation, tissue damage, and the perpetuation of inflammation in the gastrointestinal tract [21]. Recent studies suggest that targeting inflammasome activation and pyroptosis may serve as a potential therapeutic strategy for managing IBD [22]. Researchers are investigating pharmacological inhibitors targeting inflammasome components or downstream effectors involved in pyroptosis to mitigate inflammation and tissue damage in IBD [23].

The seeds of Lepidium apetalum Willdenow (LW), known as “Tinglizi” in China and “Jungryukza” in Korea, have traditionally been utilized in Oriental medicine to alleviate phlegm and edema [24]. Extracts from LW (LWE), containing compounds like helveticoside, linoleic acid, and olein, exhibit potential therapeutic effects. Notably, they have demonstrated the ability to inhibit IL-6-mediated skin pigmentation [25]. Our study aims to elucidate the protective mechanisms related to intestinal integrity and inflammation in IBD, with a specific focus on tight junction-mediated intestinal permeability in dextran sodium sulfate (DSS)-induced colitis.

## 2. Materials and Methods

### 2.1. Chemicals and Reagents

5-aminosalicylic acid (5-ASA), dextran sodium sulfate (DSS), and hematoxylin and eosin (H&E) solutions were procured from Sigma Aldrich (St Louis, MO, USA) and MP Biomedicals (Santa Ana, CA, USA), respectively. Enzyme-linked immunosorbent assay (ELISA) kits for mouse interleukin (IL)-6 and tumor necrosis factor (TNF)-α, as well as myeloperoxidase (MPO) activity, were obtained from eBioscience (San Diego, CA, USA) and Thermo Fisher Scientific (Waltham, MA, USA), respectively. Superoxide Dismutase activity assay kit was obtained from Cayman (Ann Arbor, MI, USA). RIPA lysis buffer and phosphatase and protease inhibitor cocktails were sourced from Millipore (Darmstadt, Germany) and Roche (Basel, Switzerland), respectively. Anti-ZO-1, anti-occludin, and anti-claudin-2 antibodies were purchased from Thermo Fisher Scientific. Anti-NLRP3, anti-ASC, anti-pro-caspase-1, anti-cleaved-caspase-1, and anti-GAPDH were purchased from Cell Signaling Technology (Danvers, MA, USA). 5-ASA served as the positive control.

### 2.2. Preparation of Lepidium apetalum Willdenow (LW) Extract

The hydroethanolic extract of LW was obtained from the National Institute for Korean Medicine Development (Gyeongsan, Republic of Korea). To prepare the extract, 1 kg of LW was subjected to reflux extraction with 10 L of 70% ethanol in water for 3 h. The extract was filtered through Whatman grade 2 filter paper (Whatman Limited, Maidstone, UK) and dried using a vacuum freeze-dryer (Ilshin Biobase, Gyeonggi-do, Republic of Korea). The resulting lyophilized LW powder (LWE) was stored at −20 °C until further use [26].

### 2.3. Chemical Profiling of LWE Using Ultra-High-Performance Liquid Chromatography–Tandem Mass Spectrometry (UPLC-MS/MS)

To identify the constituents of LWE, we performed ultra-high-performance UPLC-MS/MS analysis, following established protocols [27]. We used a Dionex UltiMate 3000 system paired with a Thermo Q-Exactive mass spectrometer. Chromatographic separation was achieved with an Acquity BEH C18 column (100 × 2.1 mm, 1.7 μm) using 0.1% formic acid in water and acetonitrile. Data were acquired and analyzed using Xcalibur and TraceFinder 5.1 software (Thermo Fisher Scientific, Waltham, MA, USA). Reference standards such as catechin, vicenin-2, sinapine, gentiopicroside, isoquercitrin, astragalin, sophoricoside, scoparone, quercetin, kaempferol, and isorhamnetin were procured from Targetmol (Wellesley Hills, MA, USA).

### 2.4. In Vitro Study

#### 2.4.1. Cell Culture and Viability Assay

For cell studies, Caco-2 cells (ATCC, Manassas, VA, USA) were cultured in DMEM (HyClone, Logan, UT, USA) with 1% antibiotics (HyClone) at 37 °C in a 5% CO_2_ atmosphere. The cells were seeded at a density of 1 × 10^5^ cells/well, pretreated with various doses of LW for 1 h, and then treated with 50 ng/mL IL-6 for an additional 24 h. Cell viability was measured using the Cell Counting Kit-8 (Dojindo Molecular Technologies Inc., Rockville, MD, USA). Absorbance was read at 450 nm with a VersaMax microplate reader (Molecular Devices, Sunnyvale, CA, USA). Cell viability was calculated as the absorbance of the treated sample divided by the absorbance of the vehicle control (%) [28].

#### 2.4.2. Measurement of Transepithelial Electrical Resistance (TEER) and Epithelial Paracellular Permeability

Caco-2 cells were seeded at a density of 1 × 10^5^ cells/insert in polyethylene terephthalate hanging cell culture inserts with a pore size of 0.4 µm in a 24-well plate (Millipore, Bedford, MA, USA). The medium was refreshed every 2–3 days for 21 days to allow complete differentiation. The cells were pretreated with varying concentrations of LW for 1 h and subsequently treated with 50 ng/mL IL-6. Electrical resistance was measured in triplicate using a Millicell ERS-2 voltohmmeter (Millipore), with TEER assessed after 24 h of treatment and expressed as Ohm·cm^2^. Paracellular permeability was measured using a non-absorbable FITC-conjugated dextran probe (FD-4). After 24 h of IL-6 and/or LW treatment, 1 mg/mL FD-4 was added to the apical side, PBS to the basolateral side, and the cells were incubated for 1 h at 37 °C. Absorbance on the basolateral side was measured at excitation and emission wavelengths of 490 and 520 nm using a VersaMax microplate reader [29].

#### 2.4.3. Western Blot Analysis

Caco-2 cells were homogenized in RIPA lysis buffer with added phosphatase and protease inhibitor. Protein content was quantified using a BCA assay kit (Bicinchoninic Acid, Waltham, MA, USA). Western blotting followed established protocols, allowing detailed protein analysis [28].

### 2.5. In Vivo Study

#### 2.5.1. Animal Group and Administrations

All animal experiments were approved by the Institutional Animal Care and Use Committee of the Korea Institute of Oriental Medicine (KIOM-23-067) and followed their guidelines. Six-week-old male C57/BL6 mice were obtained from DooYeol Biotech (Seoul, Republic of Korea) and assigned to five groups: Control group (Con, n = 10), 5% DSS (DSS, n = 10), 5% DSS + LW Low (LWL, n = 10), 5% DSS + LW High (LWH, n = 10), and 5% DSS + 100 mg/kg 5-ASA, as positive control (ASA, n = 10). Colitis was induced by administering 5% (wt/vol) DSS (MP Biomedicals, molecular weight 36,000–50,000) in drinking water for 5 days, followed by 3 days of DSS-free water [9]. From the first day of DSS administration, LW Low, LW High, or 5-ASA was orally given at the specific dose daily, with mice weights recorded before administration. Post-experiment, the mice were sacrificed, and colon length was measured and photographed for analysis [28,29].

#### 2.5.2. Evaluation of the Disease Activity Index (DAI)

The body weight, stool condition, and presence of gross bleeding in the mice were recorded daily to monitor disease severity throughout the experimental schedule (see Table 1). The disease activity index (DAI) was calculated as the average of the scores for [weight loss + stool condition + gross bleeding]/3 [30].

#### 2.5.3. Quantification of In Vivo Epithelial Paracellular Permeability

On day 8, paracellular epithelial permeability was assessed using FD-4. Mice were administered a 60 mg/100 g FD-4 solution. After 4 h, serum samples were collected, and FD-4 content was measured at 490 nm excitation and 520 nm emission using a VersaMax microplate reader [28,31].

#### 2.5.4. Large-Intestine Endoscopy and Histological Analysis

On the same day, high-resolution images of the colons of the anesthetized mice were captured using a mini-endoscope equipped with a visible light source (OLYMPUS, Tokyo, Japan; 670 mm length, 2.8 mm diameter). Post-endoscopy, the mice were euthanized, and whole blood and intestinal tissue were collected. Histopathological analysis was then conducted on colon tissue sections stained with Hematoxylin and Eosin (H&E), Periodic Acid-Schiff (PAS), and Masson’s trichrome solution [28,29].

#### 2.5.5. ELISA for Activity of Myeloperoxidase (MPO) and Superoxide Dismutase (SOD) and Levels of Interleukin (IL)-6 and Tumor Necrosis Factor (TNF)-α

Serum levels of IL-6 and TNF-α were measured using ELISA kits according to the manufacturer’s protocols. Additionally, MPO and SOD activity was assessed in homogenized colon tissue using MPO and Superoxide Dismutase activity assay kit [28,31].

#### 2.5.6. Real-Time PCR for Colon Tissue

Colonic tissues were promptly frozen in liquid nitrogen and homogenized with Trizol (Life Technologies, Carlsbad, CA, USA) to isolate the total RNA. The cDNA was synthesized using a RevertAid RT Reverse Transcription Kit (Thermo Fisher Scientific, Waltham, MA, USA). Quantitative real-time PCR (qRT-PCR) was preformed using the FastSYBR mixture (CWBIO) with specific primers, and expression levels were normalized to the GAPDH reference gene [31].

#### 2.5.7. Western Blot Analysis

The colonic tissues were homogenized using RIPA lysis buffer supplemented with phosphatase and protease inhibitors. The protein content was quantified using a BCA assay kit (Waltham, MA, USA). Western blotting was carried out according to previously established methods [28].

### 2.6. Statistical Analysis

Statistical analysis was performed using GraphPad Prism 10 software (GraphPad Software, San Diego, CA, USA). The data are expressed as the mean ± standard error of the mean of triplicate experiments. Statistical significance was determined using ANOVA followed by Dunnett’s post hoc test, with *p*-values less than 0.05 considered statistically significant.

## 3. Results

### 3.1. Eleven Phytochemicals of LWE Were Identified

UPLC-MS/MS is a precise analytical technique widely used for chemical profiling. It stands out as one of the most accurate assays for characterizing active ingredients in herbal medicine [32]. In our research, we employed UPLC-MS/MS analysis to identify the components in LWE. Using retention time (Rt) and mass spectral data from reference standards and previous studies (Table 2 and Figure 1), we detected several phytochemicals in LWE, including catechin (No. 1), vicenin-2 (No. 2), sinapine (No. 3), gentiopicroside (No. 4), isoquercitrin (No. 5), astragalin (No. 6), sophoricoside (No. 7), scoparone (No. 8), quercetin (No. 9), kaempferol (No. 10), and isorhamnetin (No. 11). We carefully compared these identified compounds against the retention times and mass spectra of reference standards.

### 3.2. Impact of LWE on Intestinal Barrier Function in a Barrier Model In Vitro

Next, we evaluated the effect of LWE on TEER in differentiated Caco-2 cells exposed to inflammatory stimulation (Figure 2). Notably, LWE alone did not affect Caco-2 cell viability (Figure 2A). When combined with IL-6 treatment, LWE maintained comparable outcomes (Figure 2A). IL-6 exposure resulted in a 57% reduction in Caco-2 TEER from the baseline value, whereas LWE concentration dependently restored Caco-2 TEER, starting at 12.5 μg/mL (Figure 2B). To examine paracellular permeability, we measured the flux of fluorescent molecules across the Caco-2 monolayers. Our data indicated that IL-6 increased dextran permeability by 142%, while LWE reduced permeability in a concentration-dependent manner starting at 12.5 μg/mL (Figure 2C). Furthermore, LWE restored the levels of tight junction (TJ) proteins ZO-1 and occludin in Caco-2 cells (Figure 2D–F).

### 3.3. LWE Improves Clinical Symptoms of DSS-Induced Acute Colitis

To assess the therapeutic efficacy of LWE in colitis, we induced colitis in the mice by administering 5% DSS for 5 days followed by water for 3 days while simultaneously administering LWE or 5-ASA orally as a positive control (Figure 3A). The DSS-exposed group exhibited continuous body weight loss, severe diarrhea, and fecal occult blood compared to the control group. However, mice treated with 5-ASA and LWE experienced less body weight loss compared to the DSS group. Notable differences in DAI values were observed between the DSS group and the LWE and 5-ASA groups (Figure 3B,C). Moreover, the DSS group demonstrated shorter colon lengths and larger spleen sizes compared to the control group. Conversely, the 5-ASA, LWE-L, and LWE-H groups exhibited significantly longer colon lengths than the DSS group (Figure 3D,E). The LWE-treated groups also displayed reduced spleen sizes compared to the DSS group (Figure 3D). Compared to the control group, the serum FITC-dextran content significantly increased in the DSS group but decreased in the LWE-L, LWE-H, and 5-ASA treatment groups, respectively (Figure 3F). Histologically, the control group exhibited orderly arranged epithelial cells without injury or inflammatory cell infiltration (observed through endoscopy and H&E staining) (Figure 4). In contrast, the DSS group showed a significant decrease in the height and thickness of colonic villi and increased inflammatory cell infiltration in the mucous membrane (Figure 4). In the LWE groups, both the average colon villus height and thickness increased, and the number of infiltrating inflammatory cells in the mucous membrane decreased (Figure 4). Furthermore, both LWE and 5-ASA were more effective than the DSS group at reducing histopathological damage, indicating that oral administration of LWE can prevent DSS-induced colonic tissue damage by significantly reversing the increase in intestinal permeability.

### 3.4. LWE Alleviates MPO Activity and Modulates Pro-Inflammatory Cytokine Levels

We evaluated the impact of LWE on DSS-induced MPO-mediated oxidative injury in colon tissues. Our results revealed that DSS-administered mice exhibited significantly higher peroxidation activities compared to the control group (Figure 5A). However, these effects were markedly reduced in the LWE treatment groups (Figure 5A). Additionally, we measured serum levels of IL-6 and TNF-α, which were elevated in DSS-administered mice compared to the control mice (Figure 5B,C). In contrast, the serum levels and of IL-6 and TNF-α were significantly lower in those treated with LWE compared to DSS alone (Figure 5B,C). This result indicated that LWE treatment correlates with reduced oxidative stress markers (peroxidation activities) and inflammatory cytokine levels (IL-6 and TNF-α) in mice with DSS-induced colitis.

### 3.5. LWE Improves DSS-Induced Abnormal Protein and mRNA Expression of NLRP3 Inflammasome-Related Factors in the Colon

The NLRP3 inflammasome plays a crucial role in DSS-induced murine colitis [18]. LWE significantly relieved acute colitis in mice. As shown in Figure 6A, LWE inhibited protein levels of pro-caspase-1, cleaved-caspase-1 (caspase-1 p10), ASC, and NLRP3 in colon tissues, indicating its protective effects through NLRP3 downregulation (Figure 6A–E). Additionally, LWE suppressed DSS-induced mRNA levels of IL-1β, IL-6, and TNF-α (Figure 6F–H). These findings collectively suggest that LWE effectively suppresses NLRP3 inflammasome activation in vivo.

### 3.6. LWE Alleviates Oxidative Stress in DSS-Induced Acute Colitis Mice by Activating Nrf2

To assess LWE’s impact on oxidative stress and the antioxidant defense system in colon tissue, we measured activity of SOD and mRNA levels of SOD1 and NQO1. Notably, these levels were significantly lower in the DSS-administered group than in the control group (Figure 7A–C). However, mice treated with a low dose of LWE exhibited higher SOD1 and NQO1 mRNA levels compared to the DSS-treated group (Figure 7B,C). Figure 7C illustrates LWE’s effect on the Nrf2/Keap1 pathway an upstream regulator of antioxidant enzymes via Western blot analysis. The DSS-induced mice had increased colonic Keap1 protein levels and decreased p-Nrf2 protein levels compared to the control mice (Figure 7D–F). However, LWE reversed this trend, decreasing Keap1 expression and increasing p-Nrf2 levels (Figure 7D–F). 5-ASA treatment had a similar effect, upregulating p-Nrf2 and downregulating Keap1 (Figure 7D–F). Furthermore, the mice exhibited similarly high expression levels of p-Nrf2 in consistency with mRNA results in treated with a low dose of LWE (Figure 7A–C), which may be attributed to specific conditions such as individual differences in experimental animals and in gene expression.

### 3.7. LWE Restores the Intestinal Barrier Structure and Mucosal Integrity in DSS-Induced Acute Colitis Mice

We assessed histopathological changes in DSS-induced UC and observed more pronounced fibrosis severity and goblet cell loss in the DSS group compared to the control group. Figure 8A shows PAS staining revealing purple-labeled goblet cells in mouse colon tissue. The DSS-treated mice exhibited fewer goblet cells than the control mice; under LWE treatment, the number of goblet cells increased compared to the DSS group. Additionally, the degree of fibrosis significantly decreased in the LWE-treated groups (Figure 8B). We also analyzed the gene and protein expressions of ZO-1 and occludin. The results indicated a significant reduction in ZO-1 and occludin gene and protein expressions in the DSS group compared to the control group (Figure 9). However, in the LWE-treated group, both gene and protein expressions of ZO-1 and occludin were significantly higher than in the DSS group (Figure 9).

## 4. Discussion

IBD, characterized by recurrent gastrointestinal tract inflammation due to abnormal immune responses, poses treatment challenges with current drugs. These drugs often have side effects and require long treatment durations [33,34]. Traditional herbal medicine has emerged as a potential alternative, known for its preventive and therapeutic benefits across various diseases, including IBD [35,36]. One such herbal remedy is the hydroethanolic extract of Lepidium apetalum Willdenow (LWE), commonly used in Korea. LWE is recognized for its pharmacological effects, particularly its anti-inflammatory properties [24,25]. However, its impact on IBD remains uncertain. Our study aimed to assess LWE’s pharmacological effects on Interleukin (IL)-6-induced barrier dysfunction in Caco-2 cells and DSS-induced UC in mice, focusing on antioxidant effects and intestinal barrier permeability.

First, LWE effectively restored the integrity of the intestinal barrier, as evidenced by increased TEER, reduced permeability, and elevated levels of TJ proteins ZO-1 and occludin in Caco-2 cells. This restoration is crucial in preventing the translocation of luminal antigens that exacerbate inflammation. Second, we utilized a DSS-induced colitis mouse model to assess LWE’s impact on UC. This model closely mirrors the hallmark symptoms of human UC, including weight loss, severe diarrhea, bloody stool, intestinal mucosal ulceration, and colon shortening, closely resembling those observed in human UC.

We examined the effect of LWE on UC in mice. The mice were given drinking water fortified with 5% DSS. We observed that mice treated with DSS began losing weight from the fifth day, while those treated with LWE started showing significant weight loss from the eighth day. The DAI, a measure of UC severity, was notably reduced in the LWE-treated group. Additionally, LWE treatment mitigated colon shortening and spleen enlargement, common symptoms of UC. Upon conducting endoscopy and histological analysis of the colonic mucosa, we found that LWE offered protection against damage to various layers of the colon (mucosa, submucosa, muscularis, and serosa) inflicted by DSS treatment. This suggests a significant improvement in DSS-induced UC symptoms upon LWE administration. Oral administration of DSS typically results in severe colonic mucosa damage, leading to increased intestinal permeability, mucosal hypertrophy, inflammatory cell infiltration, and intense intestinal inflammation. Inflammatory cytokines and mediators such as IL-1β, IL-6, and TNF-α are pivotal in initiating and sustaining colitis [28,31,37]. Our findings indicate that the LWE-treated group exhibited a decrease in inflammatory cell infiltration and inflammatory factors compared to the DSS-treated group. This improvement is likely due to the preservation of intestinal barrier integrity facilitated by LWE administration.

The intestinal barrier, a crucial structure separating the lumen from the external environment, is maintained by TJ. These TJs consist of various transmembrane proteins linked to cytoplasmic adapters [8,38]. These adapters enable attachment to adjacent cells and control the passage of water, ions, and other substances [39]. Higher levels of TJ proteins can preserve the structural integrity of the intestinal barrier, thereby helping to prevent colitis [40]. LWE enhances the antioxidant defense system by increasing SOD activity and the expression of antioxidant enzymes [41]. By activating the Nrf2/Keap1 pathway, LWE offers robust protection against oxidative stress, significantly contributing to the progression of colitis [42,43].

Our UPLC-MS/MS analysis identified eleven phytochemicals in LWE, including catechin, vicenin-2, and quercetin. These compounds have shown potential benefits in managing colitis and alleviating oxidative stress and inflammation [44,45]. They restore TEER, reduce paracellular permeability, and improve colitis-related symptoms in animal models. These improvements include reduced body weight loss, DAI values, and spleen size, as well as improved colon length and reduced serum FITC-dextran levels. Furthermore, these phytochemicals have antioxidant properties that effectively fight the oxidative stress associated with colitis by raising GSH levels and improving SOD activity in colon tissues. They upregulate antioxidant enzymes such as SOD1 and NQO1, contributing to cellular defense against oxidative damage [46,47]. Also, these compounds possess anti-inflammatory properties, suppressing inflammatory pathways involved in colitis and inhibiting the NLRP3 inflammasome, thereby reducing pro-inflammatory cytokines like IL-1β, IL-6, and TNF-α [48]. This modulation of inflammatory responses helps to alleviate tissue damage and inflammation severity associated with colitis [49]. In summary, catechin, vicenin-2, and quercetin present promising therapeutic potential in managing colitis by addressing oxidative stress and inflammation while promoting intestinal barrier integrity [50,51]. Our results suggest that LWE improves intestinal barrier integrity by preventing damage to TJs and preserving the mucus layer. Moreover, LWE appears to reduce inflammatory cell infiltration in colon tissue and decrease the mRNA expression of inflammatory factors.

## 5. Conclusions

LWE exhibits significant therapeutic potential for treating DSS-induced colitis by restoring intestinal barrier function, lowering inflammation, reducing oxidative stress, and improving the histopathological outcomes. Further research, including to address this gap by incorporating a treatment group that receives only LWE, thereby enabling a comprehensive assessment of its in vivo effects and clinical trials, is necessary to fully establish the efficacy and safety of LWE for treating IBD.

## Figures and Tables

**Figure 1 antioxidants-13-00795-f001:**
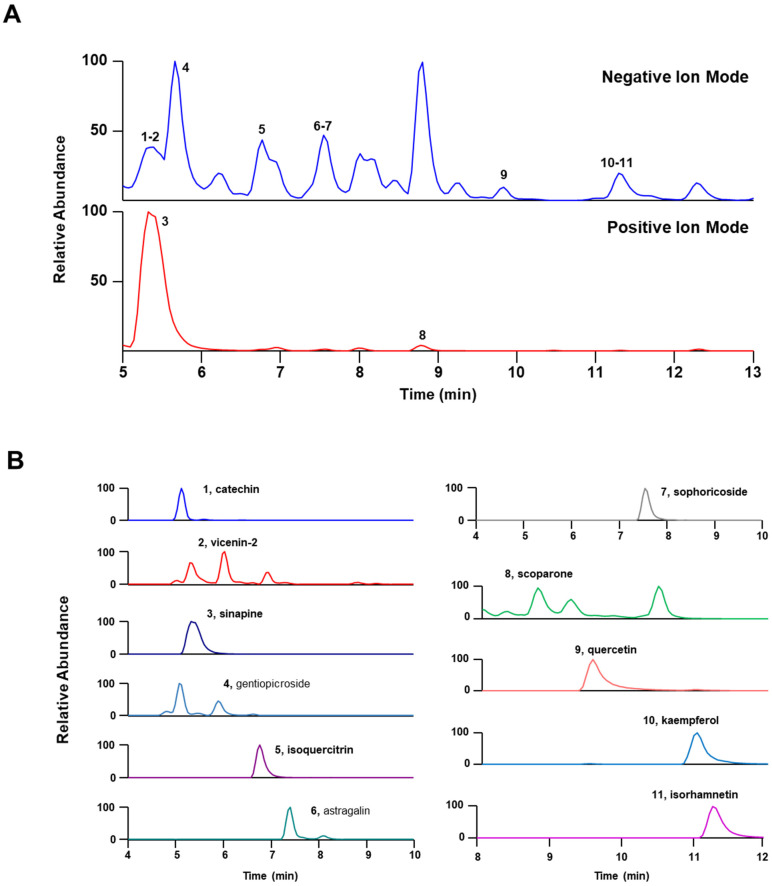
UPLC-MS/MS Analysis of the Hydroethanolic Extract of *Lepidium apetalum* Willdenow (LWE). (**A**) Base peak chromatogram in negative and positive ion mode for LWE. (**B**) Extracted ion chromatogram of LWE. The identified compounds are: 1, catechin; 2, vicenin-2; 3, sinapine; 4, gentiopicroside; 5, isoquercitrin; 6, astragalin; 7, sophoricoside; 8, scoparone; 9, quercetin; 10, kaempferol; and 11, isorhamnetin.

**Figure 2 antioxidants-13-00795-f002:**
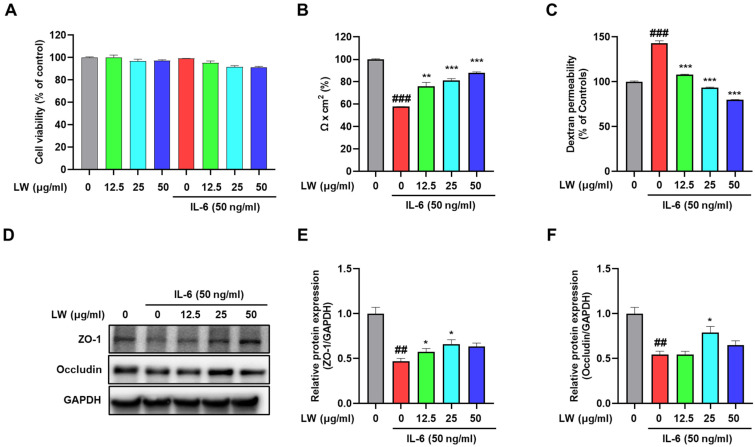
Effect of Hydroethanolic Extract of *Lepidium apetalum* Willdenow (LWE) on the intestinal barrier function of Caco-2 cells. (**A**) Cell viability post-treatment with LWE alone or in combination with IL-6. (**B**) Transepithelial electrical resistance (TEER). (**C**) Epithelial paracellular permeability. (**D**) Representative expression of zonula occludens (ZO)-1 and occludin. Densitometry for (**E**) ZO-1, (**F**) occludin. Cells were pretreated with various concentrations of LWE for 1 h followed by treatment with 50 ng/mL IL-6 for an additional 24 h. Cell viability was determined using the Cell Counting Kit-8 and is expressed as a percentage relative to the control. TEER values were used to analyze intestinal barrier integrity, and paracellular permeability was determined using a non-resorbable FITC-conjugated dextran probe (FD-4). Protein expression was measured by Western blot analysis, with GAPDH used as the protein-loading control. Results are presented as the mean ± standard error of the mean of three independent experiments vs. the Control. Significance levels are indicated as follows: ## *p* < 0.01, ### *p* < 0.001; * *p* < 0.05, ** *p* < 0.01 and *** *p* < 0.001 vs. the IL-6-treated group.

**Figure 3 antioxidants-13-00795-f003:**
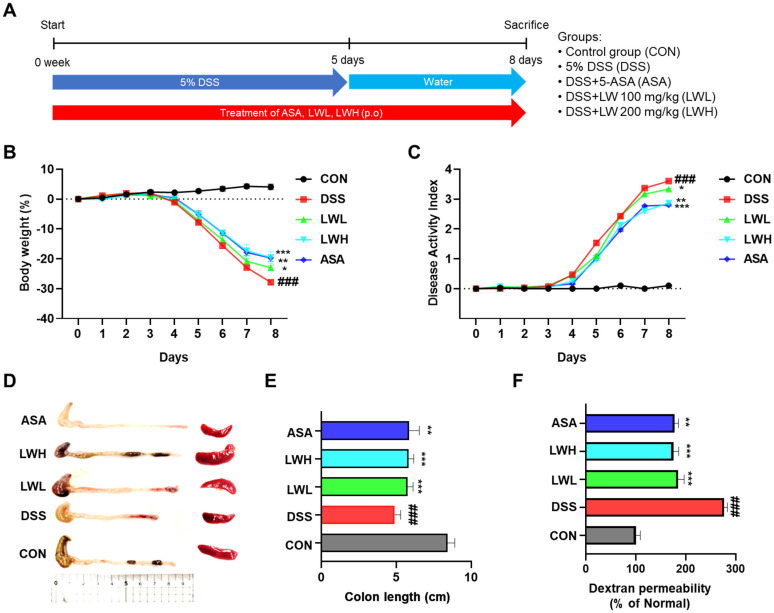
Therapeutic Effect of the Hydroethanolic Extract of *Lepidium apetalum* Willdenow (LWE) in Dextran Sodium Sulfate (DSS)-Induced acute colitis mice. (**A**) Experimental protocol for the animal study. (**B**) Body weight. (**C**) Disease activity index. (**D**,**E**) Comparison of colon length and spleen size (**F**) Serum FITC-dextran permeability. Prior to DSS treatment, mice were orally administered with LWE (100 or 200 mg/kg) or 5-aminosalicylic acid (5-ASA, 100 mg/kg). Body weight was monitored before the administration of LWE or 5-ASA during the experimental period. The length of the large intestine was measured after it was isolated from the sacrificed mouse. Epithelial paracellular permeability was measured using FD-4. Results are expressed as the mean ± standard error of the mean of each mouse in the same group. ### *p* < 0.001 vs. control group, * *p* < 0.05, ** *p* < 0.01, *** *p* < 0.001 vs. DSS treatment group.

**Figure 4 antioxidants-13-00795-f004:**
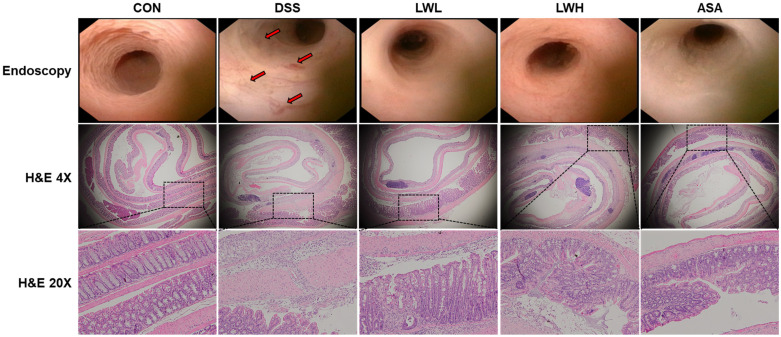
Effects of *Lepidium apetalum* Willdenow (LWE) on Histopathological Changes in Colon Tissues of Dextran Sodium Sulfate (DSS)-Induced Colitis Mice. Endoscopy and hematoxylin- and eosin (H&E)-stained images. Evaluation of DSS-induced mucosal damage using a mini-endoscope and H&E staining on day 8 of the experiment. Representative images are shown. The arrows are damage regions and squares are the enlarged part at 20× magnification. Magnification ×200.

**Figure 5 antioxidants-13-00795-f005:**
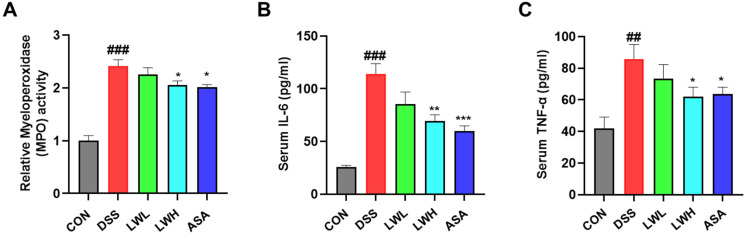
The Effects of Hydroethanolic Extract of *Lepidium apetalum* Willdenow (LWE) on Myeloperoxidase (MPO) Activities and Pro-inflammatory Cytokines Levels of Dextran Sodium Sulfate (DSS)-Induced acute colitis mice. (**A**) MPO activities in colon tissue. MPO activity was calculated using an MPO activity assay kit. Serum levels of (**B**) interleukin (IL)-6 and (**C**) tumor necrosis factor (TNF)-α as determined by enzyme-linked immunosorbent assay. Results are expressed as the mean ± standard error of the mean of each mouse in the same group, ## *p* < 0.01 and ### *p* < 0.001 vs. the control group, * *p* < 0.05, ** *p* < 0.01 and *** *p* < 0.001 vs. the DSS-treated group.

**Figure 6 antioxidants-13-00795-f006:**
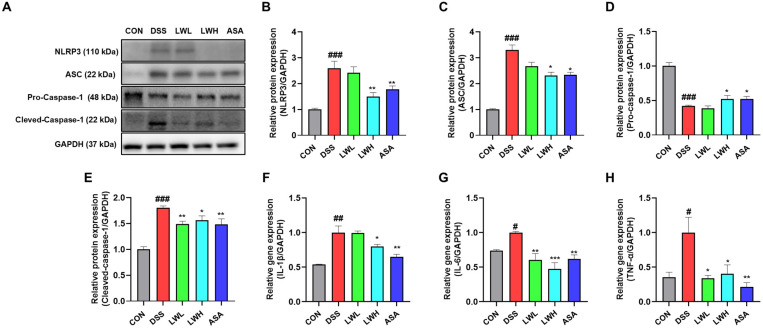
Activation of the NLRP3 Inflammasome in Dextran Sodium Sulfate (DSS)-Induced acute colitis mice by Hydroethanolic Extract of *Lepidium apetalum* Willdenow (LWE). (**A**) Representative expression of NLRP3, ASC, pro-caspase-1, and cleaved-caspase-1 expression. Densitometery for (**B**) NLRP3, (**C**) ASC, (**D**) Pro-Caspase-1, and (**E**) cleaved-Caspase-1 protein expression. Representative expression for (**F**) Interleukin (IL)-1β, (**G**) IL-6, and (**H**) tumor necrosis factor (TNF)-α mRNA expression. Results are expressed as the mean ± standard error of the mean of each mouse in the same group. Significance levels are indicated as follows: # *p* < 0.05, ## *p* < 0.01 and ### *p* < 0.001 vs. the control group, * *p* < 0.05, ** *p* < 0.01 and *** *p* < 0.001 vs. the DSS-treated group.

**Figure 7 antioxidants-13-00795-f007:**
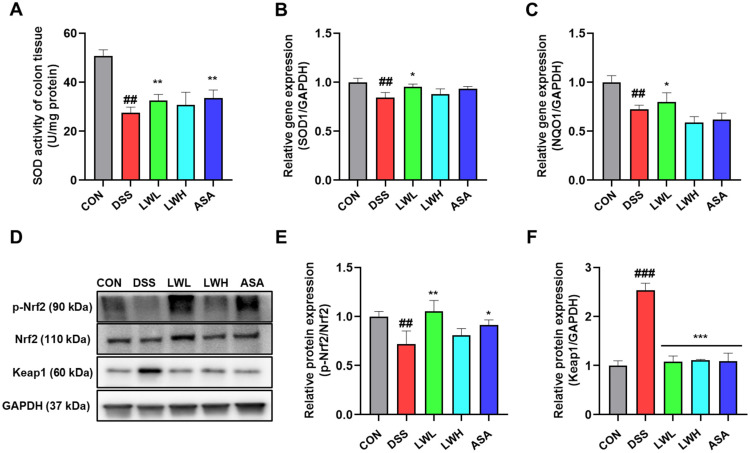
Effects of Hydroethanolic Extract of *Lepidium apetalum* Willdenow (LWE) on Oxidative Stress by Activating Nrf2 in Dextran Sodium Sulfate (DSS)-Induced acute colitis mice. Densitometery for (**A**) superoxide dismutase (SOD) activity and (**B**) SOD1 and (**C**) NAD(P)H quinone dehydrogenase (NQO) 1 mRNA expression. (**D**) Representative expression for p-Nrf2, Nrf2 and Keap1 proteins. Densitometery for (**E**) p-Nrf2/Nrf2, and (**F**) Keap1 protein expression. Results are expressed as the mean ± standard error of the mean of each mouse in the same group. Significance levels are indicated as follows: ## *p* < 0.01 and ### *p* < 0.001 vs. the control group, * *p* < 0.05, ** *p* < 0.01 and *** *p* < 0.001 vs. the DSS-treated group.

**Figure 8 antioxidants-13-00795-f008:**
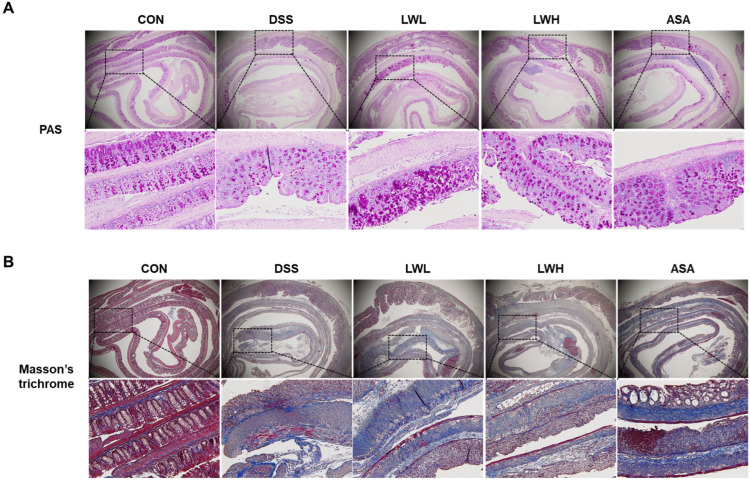
Effects of Hydroethanolic Extract of *Lepidium apetalum* Willdenow (LWE) on Intestinal Mucosal Injury Fibrosis in Dextran Sodium Sulfate (DSS)-Induced acute colitis mice. (**A**) Periodic Acid Schiff (PAS) staining for colonic goblet cells. (**B**) Masson trichromatic (MT) staining of colon tissue. Evaluation of DSS-induced mucosal damage using PAS and MT staining on day 8 of the experiment. Representative images are shown. The squares are the enlarged part at 20× magnification. Magnification ×200.

**Figure 9 antioxidants-13-00795-f009:**
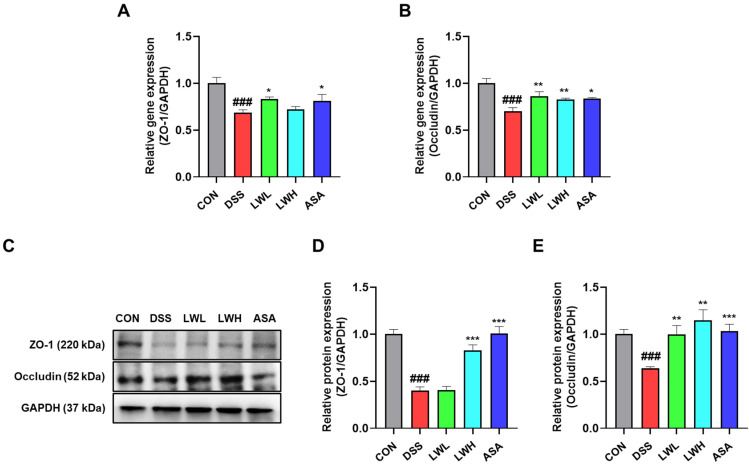
The Protective Effects of Hydroethanolic Extract of *Lepidium apetalum* Willdenow (LWE) on Tight Junctions in Mice Colon tissue of Dextran Sodium Sulfate (DSS)-induced acute colitis mice. Densitometer for (**A**) zonula occludens (ZO)-1 and (**B**) Occludin mRNA expression. (**C**) Representative expression of ZO-1 and occludin as determined by Western blot analysis. GAPDH was used as the protein-loading control. Densitometery for (**D**) ZO-1, and (**E**) Occludin. The results are presented as the mean ± standard error of the means of three independent experiments. Significance levels are indicated as follows: ### *p* < 0.001 vs. the control group; * *p* < 0.05, ** *p* < 0.01 and *** *p* < 0.001 vs. the DSS-treated group.

**Table 1 antioxidants-13-00795-t001:** Scoring System for Disease Activity Index (DAI).

DAI Score	Weight Loss (%)	Stool Condition	Rectal Bleeding
0	None	Normal	None
1	1–5		
2	5–10	Loose stools	
3	10–20		
4	>20	Diarrhea	Gross bleeding

**Table 2 antioxidants-13-00795-t002:** Characterization of Identified Compounds of Hydroethanolic Extract of *Lepidium apetalum* Willdenow (LWE) via UPLC-MS/MS analysis.

No.	Theoretical (m/z)	Measured (m/z)	Error (ppm)	Adduct	R_t_ (min)	Formula	Fragments (m/z)	Identifications
1	289.0718	289.0718	0.06	[M−H]^−^	5.12	C_15_H_14_O_6_	245, 179	Catechin
2	639.1567	639.1566	−0.1796	[M+HCO_2_]^−^	5.35	C_27_H_30_O_15_	-	Vicenin-2
3	310.1649	310.165	0.35	[M]^+^	5.37	C_16_H_24_NO_5_	251	Sinapine
4	401.1089	401.1087	−0.4883	[M+HCO_2_]^−^	5.89	C_16_H_20_O_9_	59, 89	Gentiopicroside
5	463.0882	463.0882	−0.0091	[M−H]^−^	6.77	C_21_H_20_O_12_	300	Isoquercitrin
6	447.0933	447.0932	−0.1774	[M−H]^−^	7.36	C_21_H_20_O_11_	284	Astragalin
7	477.1038	477.1034	−1.0207	[M+HCO_2_]^−^	7.55	C_21_H_20_O_10_	-	Sophoricoside
8	207.0652	207.0654	1.1575	[M+H]^+^	8.78	C_11_H_10_O_4_	119	Scoparone
9	301.0354	301.0353	−0.2364	[M−H]^−^	9.55	C_15_H_10_O_7_	151	Quercetin
10	285.0405	285.0405	0.1292	[M−H]^−^	11.01	C_15_H_10_O_6_	-	Kaempferol
11	315.051	315.051	−0.1115	[M−H]^−^	11.29	C_16_H_12_O_7_	300, 271	Isorhamnetin

Rt, retention time; All compounds were identified by comparing their Rt and mass spectra with those of genuine reference compounds.

## Data Availability

All the data supporting the results were shown in the paper, and can be obtained from the corresponding author.

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
