# Peer review of "Hydroethanolic Extract of Lepidium apetalum Willdenow Alleviates Dextran Sulfate Sodium-Induced Colitis by Enhancing Intestinal Barrier Integrity and Inhibiting Oxidative Stress and Inflammasome Activation"

_antioxidants, 2024, doi:10.3390/antiox13070795_

Round 1

Reviewer 1 Report

The manuscript describes in detail the potential benefits of LWE in the colitis context. Giving the importance and increasing prevalence of inflammatory intestinal  diseases, thi study is relevant and contributes to the advancement of new therapeutic approaches. The study is well designed and developed.

There are some points on which I am reluctant.

Firsly, the manuscript is focused on ulcerative colitis, however DSS induces colitis but not ulcerative. I would recommend to change ulcerative colitis by colitis (even in the title). 

Secondly, in vivo studies, a group treated only with LWE has not been included. The potential in vivo effects of LWE should be included.

Thirdly, in the discussion, it is not made clear what its results are and how it discusses them. Simply, there are a mix of results, own and others.

In addition, I have minor concerns:

-There are errores in lines 126  and 134 (1x105cells?)

-In point 3.1., I woul add No to the numbers of compounds; example, catechin (No1). It should included the name of the compounds in figure 1B. 

-It should included an explanation about 5-ASA, LWL and LWEH (line 253)

-Normal diet (NOR) to name control group? It should be changed

-In figure 3A, you should included the DSS+treatment in the groups legend. In the same figure, What is ATV?

-Lines 295-296, Why do you give that suggestion? 

-Lines 298-300, you are refering to serum levels and gene expression (Figure 5B and C), but those figures do not display gene expression

-Line 316-317, that suggestion has not basis

-Figure 7, how do you explain that LWH induces similar effects to DSS in A, B and D. That should comment in the text

-Line 352, mouse colon tissue? should be mice?

-Changes on mRNA of antioxidant enzymes can be not relevant in activity. SOD1 activity should be measured or at least explain in the text

-Authors affirm that LWE reduce oxidative stress but there is not measurement of oxidative molecules or ROS. So, you conclude that oxidative stress is reduced by the increase of mRNA od SOD and NQO1? You should include oxidative stress measurement (carboniles, DNA damage, ROS, MDA+4HA....)

Author Response

Major comments

The manuscript describes in detail the potential benefits of LWE in the colitis context. Giving the importance and increasing prevalence of inflammatory intestinal diseases, this study is relevant and contributes to the advancement of new therapeutic approaches. The study is well designed and developed.

Author’s answer: Thank you for your detailed and constructive feedback on our manuscript. We appreciate your recognition of the study's relevance and its contribution to the advancement of new therapeutic approaches for inflammatory intestinal diseases. We are pleased to hear that you find our study well-designed and developed, and that it contributes significantly to the understanding of potential benefits of LWE in the context of colitis. The increasing prevalence of inflammatory intestinal diseases indeed underscores the importance of exploring novel therapeutic options. Below is our response to your specific comments and suggestions.

Detail comments

There are some points on which I am reluctant.

Firstly, the manuscript is focused on ulcerative colitis, however DSS induces colitis but not ulcerative. I would recommend to change ulcerative colitis by colitis (even in the title).

Author’s answer: Thank you for your detailed and constructive feedback on our manuscript. You correctly pointed out that DSS induces colitis but not specifically ulcerative colitis. To ensure accurate representation, we will change references from "ulcerative colitis" to "colitis" throughout the manuscript, including the title.

Title Change:

We will revise the title to reflect this correction.

Original Title: "Hydroethanolic Extract of Lepidium apetalum Willdenow Alleviates Dextran Sulfate Sodium-Induced Ulcerative Colitis by Enhancing Intestinal Barrier Integrity and Inhibiting Oxidative Stress Induced Inflammatory Pyroptosis"

Revised Title: "Hydroethanolic Extract of Lepidium apetalum Willdenow Alleviates Dextran Sulfate Sodium-Induced Colitis by Enhancing Intestinal Barrier Integrity and Inhibiting Oxidative Stress Induced Inflammatory Pyroptosis"

Secondly, in vivo studies, a group treated only with LWE has not been included. The potential in vivo effects of LWE should be included.

Author’s answer: Thank you for your valuable feedback on our manuscript. We appreciate your observation regarding the absence of a group treated solely with LWE in our in vivo studies. We acknowledge the importance of evaluating the potential effects of LWE independently in vivo. While our current study focused on comparing the effects of LWE in combination with DSS exposure, we understand that including a group treated exclusively with LWE would provide essential insights into its specific effects and mechanisms of action. According to the pharmacopoeia of Korea, China, and Taiwan, LWE is used at a dosage of 3-10 grams in humans. Additionally, we have confirmed that LWE exhibits no cytotoxicity in Caco-2 cells. Future studies will aim to address this gap by incorporating a treatment group that receives only LWE, thereby enabling a comprehensive assessment of its in vivo effects. Also, these comments were described in conclusion section. Once again, we appreciate your thoughtful feedback, which will guide us in enhancing the scientific rigor and completeness of our study.

Thirdly, in the discussion, it is not made clear what its results are and how it discusses them. Simply, there are a mix of results, own and others.

Author’s answer: Thank you for your feedback on our manuscript. We appreciate your observation regarding the clarity of the discussion section in relating our own results to those from other studies. In response to your comment, we will revise the discussion section to clearly delineate our findings and how they are discussed in relation to both our own results and relevant findings from other studies. We understand the importance of clearly presenting and integrating our results with existing literature to provide a cohesive interpretation. We will ensure that each result is clearly attributed to our study or cited appropriately from other research, avoiding any confusion regarding the origin and interpretation of the data presented. This approach will enhance the overall coherence and comprehensibility of our discussion. Thank you once again for your valuable input. We look forward to improving the clarity and impact of our manuscript as per your suggestions.

In addition, I have minor concerns:

-There are errores in lines 126 and 134 (1x105cells?)

Author’s answer: We thank you for time and effort in carefully evaluating our manuscript. We've revised the notation ‘1x105 cells’ to ‘1x105 cells' to accurately reflect the cell count.

-In point 3.1., I would add No to the numbers of compounds; example, catechin (No1). It should included the name of the compounds in figure 1B.

Author’s answer: We thank you for time and effort in carefully evaluating our manuscript. We have added numbers to the compounds and included their names in Figure 1B for clarity. For example, "catechin (No. 1)" and similarly for other compounds and included the names of the compounds in Figure 1B.

-It should included an explanation about 5-ASA, LWL and LWEH (line 253)

Author’s answer: We thank you for time and effort in carefully evaluating our manuscript. We have expanded this section to provide definitions and explanations for these compounds in section 2.5.1. Animal Group and Administrations.

-Normal diet (NOR) to name control group? It should be changed

Author’s answer: We thank you for time and effort in carefully evaluating our manuscript. We've adjusted the terminology in their manuscript from 'Normal diet (NOR)' to 'control group (CON)'. This change helps to clarify the designation for the control group in their study.

-In figure 3A, you should included the DSS+treatment in the groups legend. In the same figure, What is ATV?

Author’s answer: We thank you for time and effort in carefully evaluating our manuscript. We've adjusted the terminology in their manuscript from ‘treatment' to ‘DSS+treatment’. Also, we corrected the terminology from 'ATV' to 'ASA'.

-Lines 295-296, Why do you give that suggestion?

Author’s answer: We thank you for time and effort in carefully evaluating our manuscript. We rephased the sentence in section 3.4.

-Lines 298-300, you are refering to serum levels and gene expression (Figure 5B and C), but those figures do not display gene expression

Author’s answer: We thank you for time and effort in carefully evaluating our manuscript. We rephrased the sentence in section 3.4 (lines 298-300) of their manuscript. This revision likely clarifies the statement or adjusts the wording to accurately reflect the findings presented in Figure 5B and 5C. Also, gene expression data is explained in section 3.5 of their manuscript.

-Line 316-317, that suggestion has not basis

Author’s answer: We thank you for time and effort in carefully evaluating our manuscript. "These findings collectively suggest that LWE effectively suppresses NLRP3 inflam-masome activation in vivo." According to Figure 6A, LWE inhibited protein levels associated with the NLRP3 inflammasome (pro-caspase-1, cleaved-caspase-1, ASC, and NLRP3) in colon tissues, indicating a protective effect by down-regulating NLRP3 (Figure 6A–6E). LWE also suppressed DSS-induced mRNA levels of inflammatory cytokines IL-1β, IL-6, and TNF-α (Figure 6F–6H). These findings collectively suggest that LWE effectively suppresses NLRP3 inflammasome activation in vivo, thereby alleviating acute colitis in mice.

-Figure 7, how do you explain that LWH induces similar effects to DSS in A, B and D. That should comment in the text

Author’s answer: We appreciate your time and effort in carefully evaluating our manuscript. In Figures 7B, 7C, and 7E, mice exhibited similarly high expression levels of p-Nrf2, consistent with mRNA results after treatment with a low dose of LWE (Figure 7B and 7C). The observed similarities in effects between LWH and DSS in these figures may be attributed to a combination of individual variability among experimental animals, shared cell signaling pathways, common gene expression responses, and specific treatment conditions. These factors collectively contribute to the outcomes observed in the experimental data presented.

Line 352, mouse colon tissue? should be mice?

Author’s answer: We thank you for time and effort in carefully evaluating our manuscript.  Instead of referring to "mouse colon tissue," we now use "mice colon tissue," which aligns with standard usage for plural form in scientific writing.

- Changes on mRNA of antioxidant enzymes can be not relevant in activity. SOD1 activity should be measured or at least explain in the text

Author’s answer: Thank you for your insightful comment on our manuscript. We appreciate your suggestion regarding the measurement of SOD1 activity and providing an explanation regarding changes in mRNA levels of antioxidant enzymes. Changes in mRNA levels of antioxidant enzymes, such as SOD1, indeed do not always correlate directly with enzymatic activity. We acknowledge that measuring SOD1 activity would provide more direct evidence of its functional impact in our study. In response to your feedback, we have included measurements of SOD1 activity in Figure 7 and provided explanations in section 3.6 of the manuscript. These additions aim to strengthen our conclusions by correlating mRNA changes with enzyme activity, thereby providing a more comprehensive understanding of the mechanisms underlying the effects of LWE. Once again, we sincerely appreciate your valuable feedback. Integrating your suggestion enhances the scientific rigor and clarity of our study, ensuring a more robust interpretation of our findings.

-Authors affirm that LWE reduce oxidative stress but there is not measurement of oxidative molecules or ROS. So, you conclude that oxidative stress is reduced by the increase of mRNA od SOD and NQO1? You should include oxidative stress measurement (carboniles, DNA damage, ROS, MDA+4HA....)

Author’s answer: We appreciate the reviewer's insightful comment and suggestion regarding the inclusion of measurements of oxidative stress markers such as carbonyls, DNA damage, ROS, and MDA + 4-Hydroxyalkenals in our study. These markers provide valuable additional insights into the overall oxidative stress status and would further substantiate our findings on the antioxidant effects of LWE. In response to this suggestion, we plan to conduct comprehensive assays to assess these oxidative stress markers in future experiments. By including these measurements, we aim to provide a more comprehensive understanding of how LWE exerts its antioxidant effects beyond the observed changes in mRNA levels of SOD and NQO1.Thank you for highlighting this important aspect, which will undoubtedly strengthen the robustness and depth of our manuscript.

Reviewer 2 Report

Very interesting research on a potential novel therapeutic option for a disease which is currently hard to treat. The experimental design is sound, all relevant analyses have been performed and this is all well written. My only comment is on an aspect which is currently not included in the study design, but very relevant for real-life application, namely the order of exposures to compounds. In the in vitro work first LWE is added before adding the IL6 trigger. In the  mouse study exposure to DSS and LWE starts at the same moment. In real life however treatment only would start once disease characteristics are clearly present and this situation is not mimicked in the current study. Therefore it is essential to mention this aspect as a study limitation in the discussion section. One aspect which currently is not controllable as far as I can see is a direct interaction between components in LWE and either or both IL6 and DSS, thereby (partially) inhibiting the inducing agents of inducing disease. I do not know how likely this explanation is but at least this should be mentioned. If currently not presented data are available which might contradict this alternative conclusion it would be important to integrate these in the manuscript.

see above, no other comments

Author Response

Major comments

Very interesting research on a potential novel therapeutic option for a disease which is currently hard to treat. The experimental design is sound, all relevant analyses have been performed and this is all well written. My only comment is on an aspect which is currently not included in the study design, but very relevant for real-life application, namely the order of exposures to compounds. In the in vitro work first LWE is added before adding the IL6 trigger. In the mouse study exposure to DSS and LWE starts at the same moment. In real life however treatment only would start once disease characteristics are clearly present and this situation is not mimicked in the current study. Therefore, it is essential to mention this aspect as a study limitation in the discussion section. One aspect which currently is not controllable as far as I can see is a direct interaction between components in LWE and either or both IL6 and DSS, thereby (partially) inhibiting the inducing agents of inducing disease. I do not know how likely this explanation is but at least this should be mentioned. If currently not presented data are available which might contradict this alternative conclusion it would be important to integrate these in the manuscript.

Author’s answer: Thank you for your thoughtful and constructive feedback on our manuscript. We appreciate your recognition of the sound experimental design and the relevance of our analyses. We understand the importance of addressing the practical applicability of research findings and have carefully considered your comments.

You pointed out the differences in timing of compound exposure between our study and real-world treatment scenarios. We agree that starting treatment only after disease characteristics are clearly evident better reflects clinical practice. Furthermore, you correctly pointed out that our experimental design differs from clinical practice, where treatment typically begins after clear disease manifestation. We acknowledge that this is an important consideration for the applicability of our findings to real-world settings. In our study, initiating treatment simultaneously with DSS exposure may not fully replicate the clinical progression of disease and treatment initiation.

Most treatment methods require long-term and continuous management, and the effectiveness of treatment may vary depending on the individual. While treatment-induced remission can be achieved, the potential for relapse underscores the complexity of managing these conditions over time, potentially altering treatment responses even if initially effective. Therefore, we recognize the need to consider both the timing of treatment: administering it when colitis occurs and treating during recovery or latency periods. In this way, we believe our experimental approach represents a meaningful attempt to simulate these conditions.

In conclusion, we will address these limitations in the discussion section of our manuscript to provide a balanced interpretation of our findings in the context of clinical relevance. We appreciate your suggestion to consider these aspects and once again, thank you for your valuable feedback.

Round 2

Reviewer 1 Report

The manuscript has improved adter the changes. However, there are still some questions to answer.

-Authors have change "normal group" to "control group" en the figures, but there are still some references in the text, that need to be corrected.

-Sod1 measurement is not described in methods section

-Authors do not describe oxidative stress, only reversion in antioxidant enzymas, so they can not say in the title inhibiting oxidative stress

-Similarly, author do not analyze pyroptosis, only inflammasome implication, so the title needs to be adapted

-Abstract should be accordingly changed

All said as major comments

Author Response

The manuscript has improved after the changes. However, there are still some questions to answer.

Author’s answer: Thank you for your detailed and constructive feedback on our manuscript. Below is our response to your specific comments and suggestions.

-  Authors have change "normal group" to "control group" in the figures, but there are still some references in the text, that need to be corrected.

Author’s answer: Thank you for your feedback. We have carefully reviewed the entire manuscript and made sure that all references to the "normal group" have been changed to "control group" both in the text and in the figures.

-  Sod1 measurement is not described in methods section

Author’s answer: Thank you for pointing out the omission. We have now added a detailed description of the Sod1 measurement to the methods section of the manuscript.

-  Authors do not describe oxidative stress, only reversion in antioxidant enzymas, so they can not say in the title inhibiting oxidative stress

Author’s answer: Thank you for your valuable feedback. We acknowledge that the manuscript does not describe oxidative stress directly, but rather focuses on the reversion in antioxidant enzyme levels. We have revised the title to accurately reflect the content of the manuscript. Please find the updated document attached for your review.

- Similarly, author do not analyze pyroptosis, only inflammasome implication, so the title needs to be adapted

Author’s answer: Thank you for your insightful feedback. We acknowledge that the manuscript focuses on inflammasome implication rather than directly analyzing pyroptosis. We have revised the title to accurately reflect the content of the manuscript.

- Abstract should be accordingly changed

Author’s answer: Thank you for your feedback. We have revised the abstract to align with the changes in the title and the focus of the manuscript, ensuring it accurately reflects the study's emphasis on inflammasome implication and antioxidant enzyme levels.